# Catalytic asymmetric synthesis of planar-chiral dianthranilides via (Dynamic) kinetic resolution

Chun-Yan Guan[1], Shuai Zou[1], Can Luo[1], Zhen-Yu Li[1], Mingjie Huang[1], Lihua Huang[1,2] ✉, Xiao Xiao [3], Donghui Wei [1], Min-Can Wang[1] & Guang-Jian Mei [1,2] ✉

Chirality constitutes an inherent attribute of nature. The catalytic asymmetric synthesis of molecules with central, axial, and helical chirality is a topic of intense interest and is becoming a mature field of research. However, due to the difficulty in synthesis and the lack of a prototype, less attention has been given to planar chirality arising from the destruction of symmetry on a single planar ring. Herein, we report the catalytic asymmetric synthesis of planar-chiral dianthranilides, a unique class of tub-shaped eight-membered cyclic dilactams. This protocol is enabled by cinchona alkaloid-catalyzed (dynamic) kinetic resolution. Under mild conditions, various $C_2$- or $C_1$-symmetric planar-chiral dianthranilides have been readily prepared in high yields with excellent enantioselectivity. These dianthranilides can serve as an addition to the family of planar-chiral molecules. Its synthetic value has been demonstrated by kinetic resolution of racemic amines via acyl transfer, enantiodivergent synthesis of the natural product eupolyphagin, and preliminary antitumor activity studies.

Chirality, an inherent geometric property of any three-dimensional object that cannot coincide with its mirror image, plays a crucial role in a range of disciplines including chemistry, medicine, materials, and life sciences (Fig. 1A)[1]. Central chirality, which is based on sp[3] hybridized stereogenic centers (X = C, Si, P, S, etc.) with four different groups, constitutes the most common one. Conformational chirality, in addition, arises when a particular molecular conformation is sufficiently stable. For instance, the restricted rotation of a $\sigma$-bond leads to axial chirality[2–4], and steric repulsion between the terminal aromatic rings of helicenoids results in helical chirality[5,6]. Currently, the catalytic asymmetric synthesis of molecules with central, axial, and helical chirality is a topic of intense interest and is becoming a mature field of research. In sharp contrast, planar chirality arising from the destruction of symmetry on a single planar ring has received less attention[7]. Metallocenes[8–13], cyclophanes[14–21], and some rigid cycloalkenes[22–28] are

typical planar-chiral structures. With recognition of the importance of planar chirality, particularly in chiral ligand/catalyst discovery[29–33], several catalytic enantioselective synthesis methods have been developed[34–39]. This is the case for assembling planar-chiral ferrocenes via enantioselective C−H activation by You, Zhou, and others[40–45]. Despite these achievements, difficulties in synthesis and the lack of prototypes have greatly limited the application of planar chirality, and the catalytic asymmetric synthesis of planar-chiral compounds is still in its infancy.

Dianthranilides, dibenzo[$b$,$f$][1,5]diazocine-diones, are a unique class of eight-membered cyclic lactams (Fig. 1B)[46]. The secondary cis-amide groups and the particular tricyclic ring system make dianthranilide an attractive scaffold for use in supermolecule, materials chemistry, and pharmaceuticals[47–49]. For example, the natural alkaloid eupolyphagin exhibited a promising cytotoxic effect on cancer cell

[1]Henan Key Laboratory of Chemical Biology and Organic Chemistry, College of Chemistry, Zhengzhou University, Zhengzhou, China. [2]Pingyuan Laboratory (Zhengzhou University), Zhengzhou, China. [3]Collaborative Innovation Center of Yangtze River Delta Region Green Pharmaceuticals, Zhejiang University of Technology, Hangzhou, China. ✉e-mail: hlh606@zzu.edu.cn; meigj@zzu.edu.cn

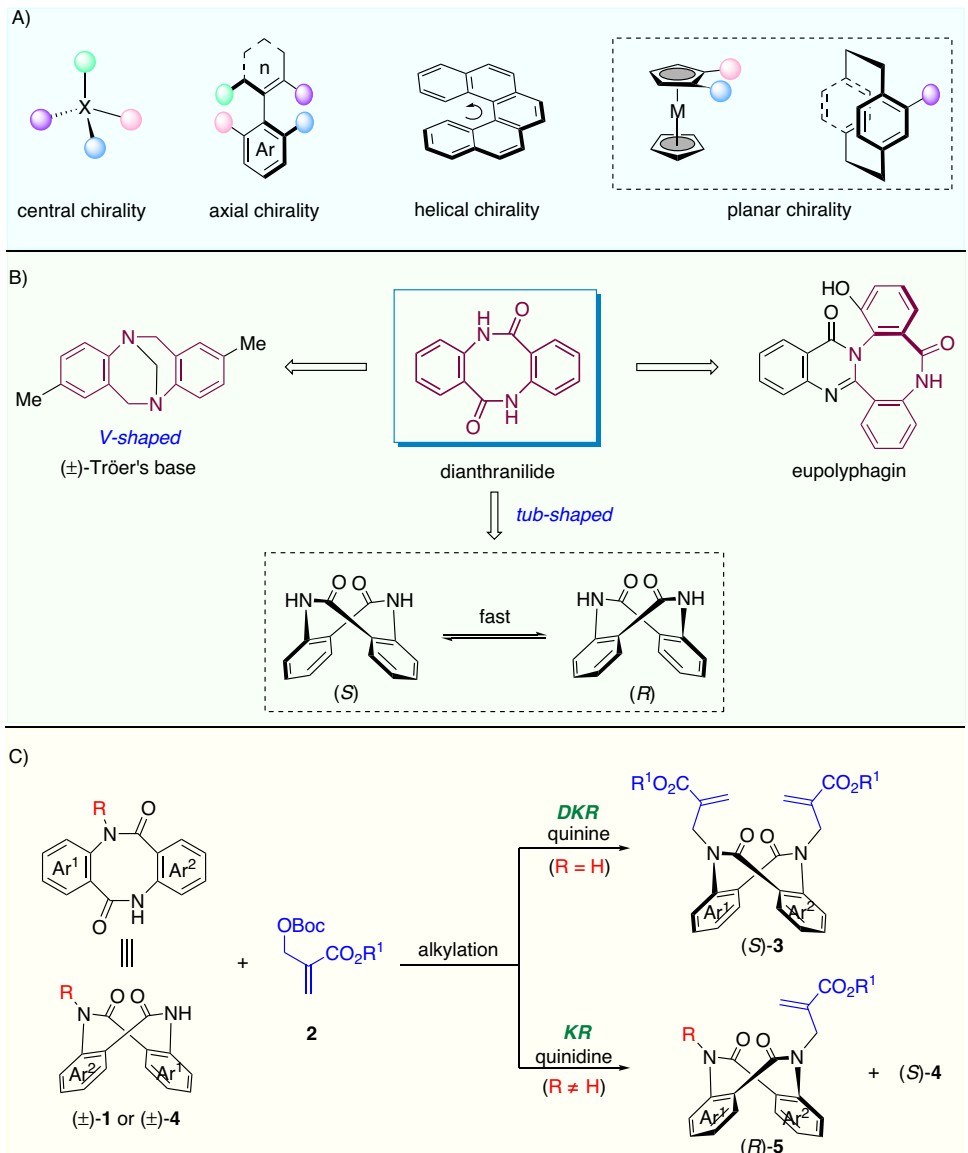

**Fig. 1 | Enantioselective synthesis of planar-chiral dianthranilides. A** Typical chiral element. **B** Planar-chiral dianthranilide. **C** DKR and KR (this work).

lines[50]. Analogous to rigid V-shaped Tröger's base in structure and geometry[51], dianthranilide adopts a tub-shaped dominant conformation and therefore is a $C_2$-symmetric planar-chiral molecule[52,53]. However, the unlocked flexible conformation allows fast interconversion of the two enantiomers (*S* and *R* forms) at ambient temperature. As a result, dianthranilides are usually racemic, and their catalytic asymmetric synthesis poses a daunting challenge. To our knowledge, in 2017, Tan et al. reported the only example of enantioselective synthesis of a dianthranilide derivative[54]. Their strategy was to construct the key axially chiral intermediate followed by the formation of an eight-member lactam via axial-to-planar chirality transfer. Given their many intriguing features and easy accessibility, dianthranilides could serve as an additional prototype of planar-chiral molecules. Herein, we set our goal to develop direct catalytic asymmetric methods to access conformationally stable planar-chiral dianthranilides with the ability to achieve chiral induction via asymmetric catalysis. To obtain rigid conformers, inhibiting boat-to-boat ring inversion is key[55,56]. We assume that N-alkylated dianthranilides **3-5** are more conformationally stable and that their catalytic asymmetric synthesis could be enabled by dynamic kinetic resolution (DKR) or kinetic resolution (KR) (Fig. 1C)[57–60].

## Results

### Reaction development

To determine its feasibility, dianthranilide **1a** was chosen as the model substrate, and Morita–Baylis–Hillman (MBH) adduct **2a** was selected as the alkylating agent. To our delight, upon the catalysis of various commercially available cinchona alkaloids, asymmetric dialkylation readily occurred, allowing the formation of conformationally stable **3a** via DKR (Table 1). In the presence of cinchonidine (**C1**), the alkylation-enabled DKR afforded product **3a** in 45% yield with 18% *ee* (entry 1). The solvent effect was then investigated, and acetonitrile was identified as the solvent of choice (entries 2–4). Consequently, catalyst screening was carried out in acetonitrile (entries 5–9). While all catalysts were effective, quinine (**C5**) and quinidine (**C6**) were found to be most effective in terms of yield and enantioselectivity (entries 8–9). Notably, quinine and quinidine afforded different enantiomers of **3a**.

### Substrate scope

With the best conditions, the substrate scope of this DKR reaction was subsequently studied (Fig. 2). First, the generality of the MBH adducts was tested (**3b–f**). The excellent yields and *ee* values indicated that various ester groups were well tolerated. Next, we turned our attention

## Table 1 | Reaction optimization

| Entry[a] | Cat. | Solvent | product | Yield (%)[b] | ee (%)[c] |
|---|---|---|---|---|---|
| 1 | **C1** | CH$_2$Cl$_2$ | **3a** | 45 | 18 |
| 2 | **C1** | toluene | **3a** | 50 | 10 |
| 3 | **C1** | THF | **3a** | 70 | 40 |
| 4 | **C1** | CH$_3$CN | **3a** | 89 | 83 |
| 5 | **C2** | CH$_3$CN | (ent)-**3a** | 63 | 75 |
| 6 | **C3** | CH$_3$CN | **3a** | 87 | 83 |
| 7 | **C4** | CH$_3$CN | (ent)-**3a** | 80 | 90 |
| 8 | **C5** | CH$_3$CN | **3a** | 95 | 98 |
| 9 | **C6** | CH$_3$CN | (ent)-**3a** | 90 | 97 |

[a] Unless indicated otherwise, reaction conditions: **1a** (0.05 mmol), **2a** (0.12 mmol), and **Cat.** (10 mol%) in a specified solvent (1 mL) at room temperature (r.t.) for 6 h.

[b] Isolated yields.

[c] Determined by chiral HPLC analysis.

to the scope of accessing dianthranilides **1**. The reaction was applicable to a range of symmetrically substituted dianthranilides, leading to the formation of dialkylated products **3 g–v** in generally excellent yields and enantioselectivities. Neither the position (from the C3- to C6- position) nor the electron nature (electron-donating or electron-withdrawing) of the substituents had an obvious effect on this reaction. In addition, di-substituted and phenyl-fused substrates were employed, which delivered corresponding products **3w–y** in excellent yields and *ee* values. Unsymmetric dianthranilides are challenging substrates for DKR, since intriguing regioisomers may be created in the 1st alkylation, thus affecting the enantioselectivity. Remarkably, when unsymmetric dianthranilides were subjected, the reaction readily occurred, giving the non-*C$_2$*-symmetric products **3z** and **3aa**. Considering the consistently excellent enantioselectivity of both the *C$_2$*- and *C$_1$*-symmetric products, it can be concluded that this cinchona alkaloid-catalyzed alkylation had good face selectivity and was not affected by steric hindrance. The absolute configurations of these N-alkylated products were assigned on the basis of X-ray crystallographic analysis of **3 m**.

To elucidate the reaction mechanism, control and kinetic experiments were performed (Fig. 3). Under racemic conditions, enantioenriched monoalkylation product **1a'** underwent a 2$^{nd}$ N-alkylation reaction with the MBH adduct **2a** to afford product **3a** with a constant *ee* (Fig. 3A). In addition, racemization experiments of **1a**, **1a'** and **3a** were conducted to investigate their configurational stabilities (Fig. 3B). The low rotational barrier (23.3 kcal/mol) indicated a dynamic interconversion in **1a**. On the other hand, the monoalkylation product **1a'** was stable enough, with an inversion barrier of 28.2 kcal/mol, while that of dialkylation product **3a** was 32.6 kcal/mol[61]. Therefore, the monoalkylation product **1a'** should be the key intermediate, and the 1st N-alkylation is the DKR and enantio-determining

step. In addition, kinetic studies were conducted to explore the reaction pathway. As shown in Fig. 3C, the formation of dialkylation product **3a** was fast. While the *ee* of **3a** was maintained at the same high level, substrate **1a** was always racemic. This further verified the previous speculation that dialkylation is a DKR process. Furthermore, presynthesized racemic **1a'** was used under standard conditions (Fig. 3D). Enantioenriched **3a** was obtained initially, and its *ee* decreased over time. The *ee* of substrate **1a** increased over time. These results indicated a distinct KR process. To demonstrate this concept, the KR of racemic **1a'** was carried out (Fig. 3E). Under standard conditions, racemic **1a'** reacted smoothly with 0.6 equivalents (eq.) of **2a**. The corresponding **3a** was obtained in 49% yield with 95% *ee*, and **1a'** was recovered in 48% yield with 99% *ee*.

Considering that the excellent *s*-factor (*s* = 205) is comparable to that of enzyme catalysis, this KR reaction is synthetically useful. In this context, we prepared several mono-substituted dianthranilides and subjected them to the KR. The results are summarized in Fig. 4. The variation of the MBH adducts was tested. Various ester groups, such as -CO$_2$$^t$Bu, -CO$_2$Et, -CO$_2$$^n$Bu, -CO$_2$$^i$Bu, and -CO$_2$Bn, were well tolerated, affording alkylated products **5a–e** and recovered **4a** in good yields with excellent *ee*. Furthermore, dianthranilides bearing substituents on the phenyl ring were also compatible, but the enantioselectivity of products **5 f–j** decreased to some extent. The generality of the N-substituents was then investigated. In addition to the substituted benzyl groups (**5k–m**), allyl (**5n**), 2-ethoxy-2-oxoethyl (**5o**), and phenyl (**5p**) groups were also applied. Notably, dianthranilide substrates featuring acyl groups of N-substitutions had flexible conformations, thus providing DKR results (**5q & 5r**).

Based on the above experimental results and previous reports on asymmetric amide N-alkylations with MBH adducts[62–65], a plausible reaction mechanism and transition state have been proposed to

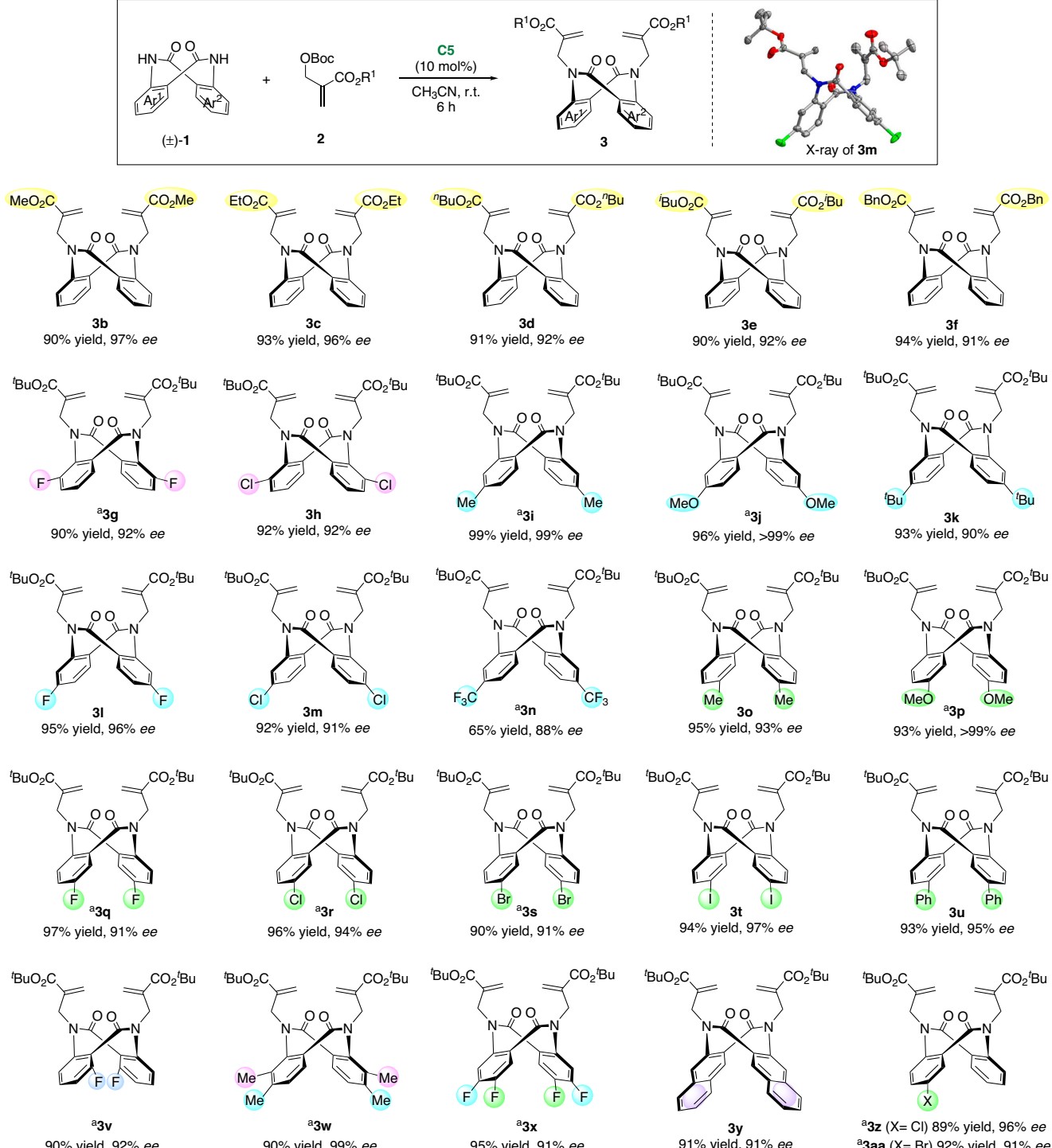

**Fig. 2 | Substrate scope of DKR.** Reaction conditions: **1** (0.1 mmol), **2** (0.24 mmol), and **C5** (10 mol%) in CH$_3$CN (1 mL) at room temperature (r.t.) for 6 h, isolated yield, *ee* was determined by chiral HPLC. [a]Using **C6**.

elucidate the origin of the high stereoselectivity of these reactions (Fig. 5). Taking the synthesis of dialkylation product (*R*)-**3a** as an example, the catalyst quinidine (**C6**) reacts with MBH adduct **2a** in an S$_N$2′ fashion to form the chiral adduct **C6′**, CO$_2$, and *tert*-butoxide anion. Then, the resulting **C6′** distinguishes the two enantiomers of substrate **1a** by hydrogen-bonding interactions. The conformationally matched (*R*)-**1a** forms a dominant hydrogen bonding network with **C6′** via transition state **TS-R**. However, the same hydrogen bonding network will cause additional steric hindrance for the mismatched (*S*)-**1a** (**TS-S**). As a result, (*R*)-**1a** undergoes asymmetric N-alkylation

accompanied by the regeneration of catalyst **C6**, while (*S*)-**1a** racemizes due to its low rotational barrier. In the case of excessive alkylation reagents, the monoalkylated product **1a′** generated in situ will rapidly undergo a second alkylation reaction, delivering dialkylation product **3a**. The preliminary DFT calculations support our hypothesis, and this transition state model is also applicable to the KR (see Supplementary Figs. 242–244 and Source Data for details).

Finally, to highlight the synthetic utility, scaled-up DKR, and KR experiments were conducted (Fig. 6A). Under standard conditions, the DKR of racemic **1a** furnished dialkylated product **3a** in

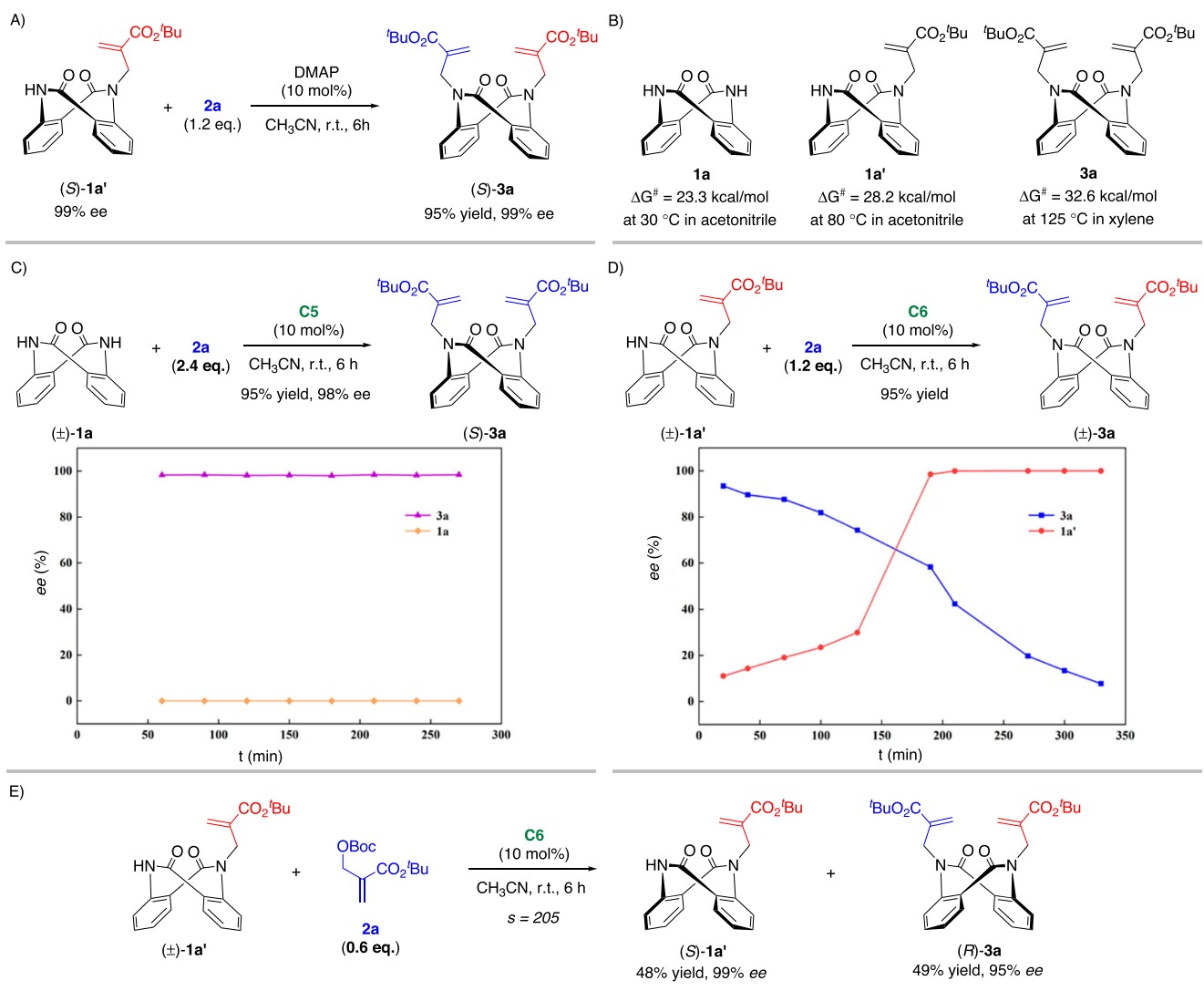

**Fig. 3 | Mechanism consideration and kinetic resolution. A** *Ee* maintenance experiment. **B** Racemization experiment. **C, D** Kinetic experiments. **E** Kinetic resolution.

95% yield with 98% *ee*. Notably, one of these two allyl groups can be selectively removed with NaOH, delivering monoalkylated intermediate **1a'** with a maintained *ee*. Further, acylation with BzCl gave product **6a**, which was proven to be a potential chiral acylation reagent in the subsequent investigations. Moreover, the KR of racemic **4a** readily occurred with a perfect *s*-factor, affording (*S*)-**4a** and the (*R*)-**5a** with excellent enantioselectivity. More importantly, the dealkylation of (*R*)-**5a** can in turn generate (*R*)-**4a**. In this way, the direct resolution of (±)-**4a** can be achieved, allowing access to its two enantiomers with a single chiral catalyst. In addition, these synthesized dianthranilides could serve as potential chiral reagents for acyl transfer (Fig. 6B). Direct acylation of **1a'** and **4a** afforded compounds **6** with maintained *ee* values. Treatment of **6** with racemic 3,3-dimethylbutan-2-amine regenerated **1a'** and **4a** with an excellent recycling rate of 95%. At the same time, the KR of (±)-**7** was achieved with moderate to good selectivity. This preliminary attempt indicated that conformationally stable planar-chiral dianthranilides could serve as a promising platform for enantioselective synthesis. To further demonstrate the practicality of this protocol, the enantiodivergent synthesis of the natural product eupolyphagin was accomplished (Fig. 6C). Starting from the commercially available 2-iodo-6-methoxyaniline **9**, racemic intermediate **10** can be obtained in 5 steps according to Tan's procedures[54]. Then, the projected KR served as the key step

to afford the highly enantioenriched precursor (*S*)-**10** and alkylation product (*R*)-**11**. Dealkylation of (*R*)-**11** led to the (*R*)-**10** precursor. Subsequent deprotection of the methyl group with BBr$_3$ afforded the natural products (+)-eupolyphagin and (-)-eupolyphagin in good yields.

Moreover, we selected a few products (**3a, 3 f, 3 g, 3 h, 3j, 3 m, 3p** and **3q** for DKR; **5a, 5e, 5 f, 5 g, 5 h, 5n** and **5o** for KR) for anticancer activity evaluation of cell viability via the CTG assay for A2780 (ovarian cancer), HeLa (cervical carcinoma), HT-29 (colon cancer), LoVo (colon cancer), MV-4-11 (acute monocytic leukemia), and U87-MG (astrocytoma) human cancer cell lines (see the Supplementary Table 16 for details). Compounds **3a** (A2780 cells, IC$_{50}$ = 13.89 µM; HeLa cells, IC$_{50}$ = 37.28 µM; HT-29 cells, IC$_{50}$ = 28.41 µM; LoVo cells, IC$_{50}$ = 36.25 µM; MV-4-11 cells, IC$_{50}$ = 21.00 µM; U87-MG cells, IC$_{50}$ = 47.68 µM), **3q** (A2780 cells, IC$_{50}$ = 16.24 µM; Hela cells, IC$_{50}$ = 61.51 µM; HT-29 cells, IC$_{50}$ = 54.07 µM; LoVo cells, IC$_{50}$ = 53.60 µM; MV-4-11 cells, IC$_{50}$ = 38.79 µM; U87-MG cells, IC$_{50}$ = 109.33 µM), and **5 g** (A2780 cells, IC$_{50}$ = 10.58 µM; HeLa cells, IC$_{50}$ = 16.80 µM; HT-29 cells, IC$_{50}$ = 15.06 µM; LoVo cells, IC$_{50}$ = 14.75 µM; MV-4-11 cells, IC$_{50}$ = 14.27 µM; U87-MG cells, IC$_{50}$ = 25.43 µM) exhibited significant and broad anticancer potency (Fig. 6D). Consequently, these planar-chiral dianthranilides have the potential to become lead anticancer compounds. Further structure-activity relationship-studies are ongoing in our laboratory.

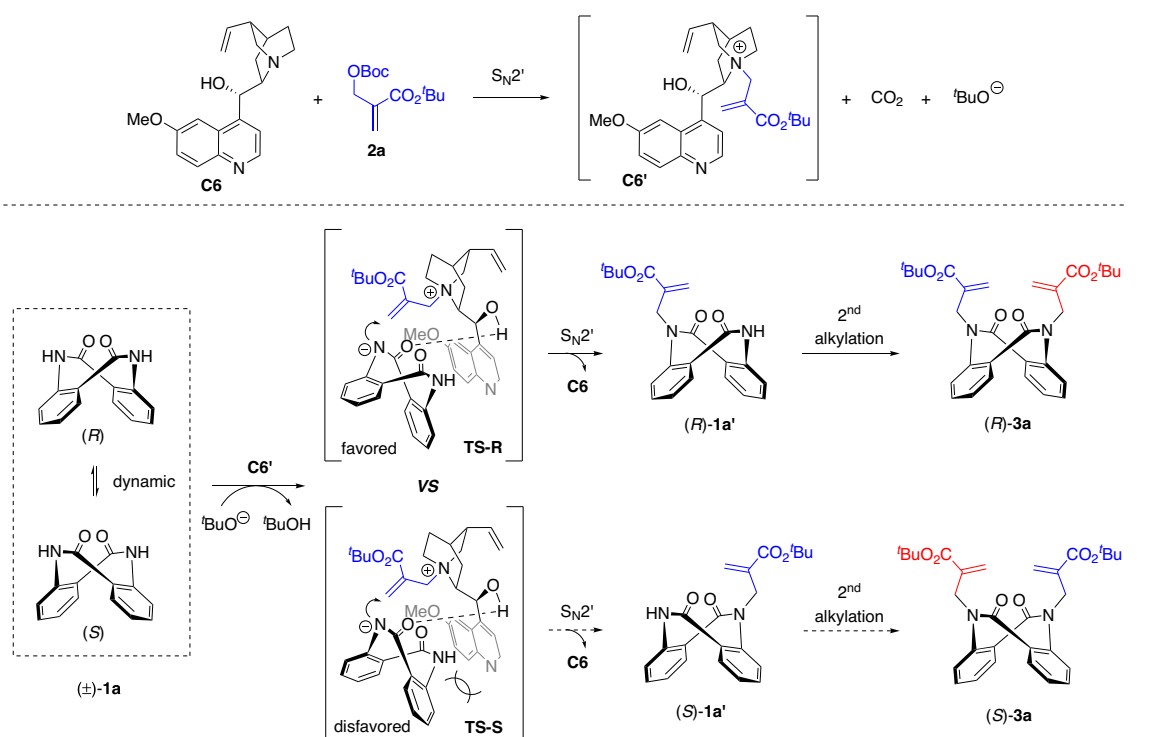

**Fig. 4 | Substrate scope of KR.** Reaction conditions: **4** (0.1 mmol), **2** (0.06 mmol), and **C6** (10 mol%) in CH₃CN (1 mL) at room temperature (r.t.) for 6 h, isolated yield, *ee* was determined by chiral HPLC. Conversion (C) = ees/(ees + eep). s = ln[(1 − C)(1 − ees)]/ln[(1 − C)(1 + ees)].

**Fig. 5 | Proposed reaction pathway and transition state.**

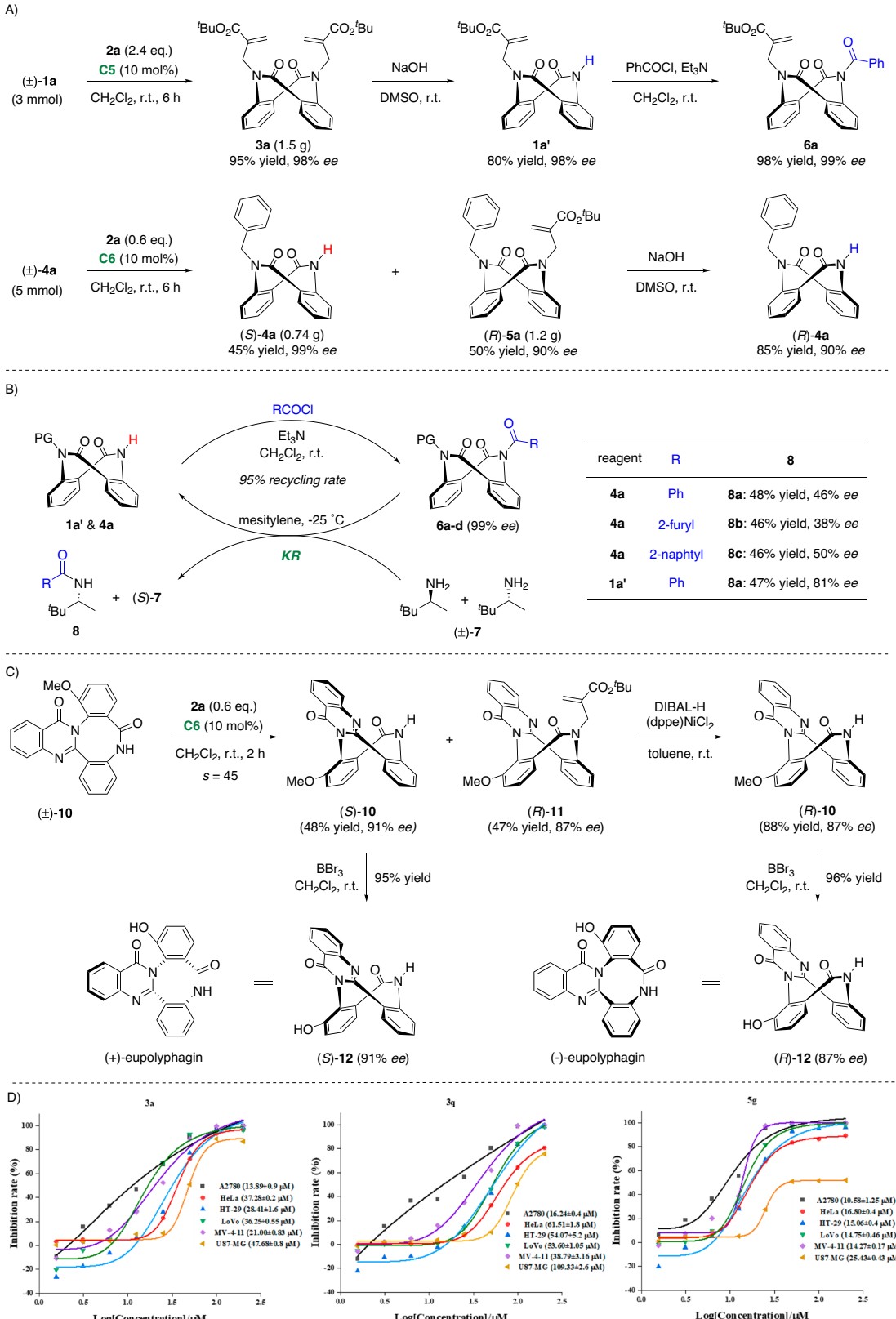

**Fig. 6 | Further elaborations. A** Large-scale synthesis and derivatization. **B** Acyl transfer. **C** Enantiodivergent synthesis of eupolyphagin. **D** Cytotoxicity of the synthesized compounds in various human cells.

In conclusion, we have accomplished the catalytic enantioselective synthesis of planar-chiral dianthranilides. In contrast to the well-studied central, axial, and helical chirality, less attention has been paid to planar chirality due to the difficulty in synthesis and the lack of a prototype. Herein, we report a highly efficient (dynamic) kinetic resolution protocol for the synthesis of tubshaped dianthranilides, which serve as an addition to the family of planar-chiral molecules. Under the catalysis of cinchona alkaloids,

various $C_2$- or $C_1$-symmetric planar-chiral dianthranilides have been readily prepared in high yields and with excellent enantioselectivities. The preliminary attempt at the kinetic resolution of racemic phenylethylamine via acyl transfer demonstrated that conformationally stable planar-chiral dianthranilides can serve as a promising platform for enantioselective synthesis. Using this method, the enantiodivergent synthesis of the natural product eupolyphagin was accomplished. Further applications and other related investigations along this line are ongoing and will be reported in due course.

## Methods

### General procedure for DKR

Substrate **1** (0.10 mmol) and catalyst **C5** (10 mol%) were dissolved in $CH_3CN$, and MBH ester **2** (0.24 mmol) was added. The reaction mixture was stirred for 6 h at room temperature. The solvent was removed in vacuo and the crude product was separated by flash column chromatography on silica gel (petroleum ether/ethyl acetate = 4:1) to afford **3**.

### General procedure for KR

Substrate **4** (0.10 mmol) and catalyst **C6** (10 mol%) were dissolved in $CH_3CN$, and MBH ester **2** (0.06 mmol) was added. The reaction mixture was stirred for 6 h at room temperature. The solvent was removed in vacuo and the crude product was separated by flash column chromatography on silica gel (petroleum ether/ethyl acetate = 3:1 – 1:1) to afford recovered (*S*)-**4** and (*R*)-**5**.

### Reporting summary

Further information on research design is available in the Nature Portfolio Reporting Summary linked to this article.

## Data availability

The authors declare that the data relating to the characterization of products, experimental protocols, and computational studies are available within the article and its Supplementary Information. The data for the crystal structure **3 m** reported in this paper were deposited at the Cambridge Crystallographic Data Center (CCDC) under the deposition number CCDC 2271258. Copies of the data can be obtained free of charge via www.ccdc.cam.ac.uk/data_request/cif. Further data supporting the findings of this study are available from the corresponding author upon request. Source data are provided with this paper.

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

## Acknowledgements

Financial support from the National Natural Science Foundation of China (22371265 and 22208302) and the Natural Science Foundation of Henan Province (222300420084) is gratefully acknowledged.

## Author contributions

C.-Y.G., C.L., Z.-Y. L., and X.X. performed and analyzed the experiments. M.-C.W. participated in the early development of the project. S.Z. and D.W. performed the DFT calculations. M.H. and L.H. performed and analyzed the anticancer tests. G.-J.M. conceived and designed the project. G.-J.M. supervised the project overall. All authors prepared this manuscript.

## Competing interests

The authors declare no competing interests.
