## [Peer Review File · Nature Communications]

Catalytic Asymmetric Synthesis of Planar-Chiral Dianthranilides via (Dynamic) Kinetic ResolutionREVIEWER COMMENTS

Reviewer #1 (Remarks to the Author):

In this manuscript, Mei and co-workers reported on the catalytic asymmetric synthesis of planar-chiral dianthranilides, a distinctive tub-shaped eight-membered cyclic dilactams, using both dynamic kinetic resolution and kinetic resolution protocols. The authors utilized cinchona alkaloid-catalyzed asymmetric N-allylation of the secondary amide group with MBH adduct, which resulted in the production of diverse C2- or C1-symmetric planar-chiral dianthranilides with high yields and excellent enantioselectivities. Moreover, the authors explore the application of these chiral products as chiral acyl-transfer reagents in the kinetic resolution of amines, albeit with only moderate selectivities. Notably, the authors have also achieved the enantioselective synthesis of a planar chiral natural product (+)-eupolyphagin, by using their developed kinetic resolution method. Overall, this manuscript represents a remarkable advances in the catalytic asymmetric synthesis of planar chiral molecules, making it a fit for a journal like Nature Communications, once the following points are addressed:

- 1) Despite the authors and previous research labeling these distinctive tub-shaped eight-membered cyclic dilactams as planar chirality, this reviewer believes this uncommon type of chirality could also be termed inherent chirality. It is recommended to refer to a recent mini-review (Eur. J. Org. Chem. 2023, e202300738) which discusses this topic.
- 2) Regarding the kinetic resolution reaction scope, the authors have only tested racemic dianthranilides with mono-N-alkylated substrates. It would be insightful to know whether the authors have evaluated racemic dianthranilide substrates featuring other types of N-substitutions, including aryl, acyl, and the like.
- 3) To elucidate the origins of the high stereoselectivity of these reactions, it would be beneficial to propose a plausible reaction mechanism and an asymmetric transition state.

Reviewer #2 (Remarks to the Author):

The manuscript by Mei and coworkers describes a highly efficient methodology to access enantioenriched dianthranilides by dynamic kinetic- and kinetic resolutions. For both approaches different C2- or C1-symmetric planar-chiral dianthranilides with high enantioselectivities are produced.

This transformation represents a welcome contribution to the poor set of catalytic methodologies that produce planar chirality, but it focuses exclusively on a restricted framework and a particular transformation; the DKR substrate scope shows up to 25 different structures but, from a synthetic and applicability point of view, many of them are redundant.

To clarify the reaction mechanism, the authors performed several control experiments. The main concern is that the explanations and their representation are not easy to follow and are not presented in sufficient detail. Moreover, they do not imply, at any point, the role of the catalyst.

On the other hand, the synthetic value of these planar chiral molecules as catalysts is exemplified in kinetic resolutions of secondary amines but the results provided are rather moderate.

Finally, enantioenriched(+)-eupolyphagin is prepared. The evaluation given by the authors

on page 2, of the synthetic route described earlier, is not too far from the one described by the authors themselves as a similar number of steps is needed and the enantioenriched product is obtained by kinetic resolution.

Reviewer #3 (Remarks to the Author):

The paper by Guan et al. presents a groundbreaking approach to the dynamic kinetic resolution (DKR) of planar chiral dianthranilides using quinine/quinidine-catalyzed N-alkylation with allylic carbonates. The resolution achieved high efficiency, yielding dialkylated products with significant enantiomeric excess (ee). The study extensively explored the substrate scope for both dianthranilides and allylic carbonates, including successful outcomes with unsymmetrical dianthranilides. Additionally, the highly efficient kinetic resolution of racemic N-monoalkylated dianthranilides was observed. The high ee product – in fact, it was an unreacted enantiomer of the substrate – was successfully served as a chiral auxiliary for the kinetic resolution of chiral amines. Moreover, one of the compounds underwent direct derivatization into a natural product.

While the paper undoubtedly advances the field of planar chiral compound preparation, the reviewer expresses reservations about its suitability for publication in *Nature Communications*. The reviewer argues that the paper lacks a truly novel scientific concept and its impact may be limited to the chemistry community, suggesting that a chemistry-specific journal like *JACS*, *Angew. Chem.*, or *Chem. Sci.* might be more appropriate for publication.

The reviewer raises several important points for improvement:

1. Detailed Experimental Information in Supporting Information: The reviewer requests more comprehensive experimental details and results in the supporting information. Specifically, in case the substrate for (D)KR is a new compound, additional data for them, including chemical yield, product weight, physical properties, melting point, and elemental analysis or high-resolution mass spectrometry (HRMS) data, are needed. Notably, information on unsymmetrical substrates (1x and 1y) is entirely missing, and clarification is needed on whether these compounds were prepared using the same procedure as 1a. Additionally, for compounds 4k, 4l, 4m, 4n, and 4o, the reviewer highlights a lack of clarity on their synthesis process despite the provided chemical yield and spectral data. Preparation method for them, along with details on purification processes, should be provided.
2. Demonstration of Product Usage: The reviewer notes that while the paper successfully demonstrates the use of non-reacted, recovered starting compounds for KR, it falls short in illustrating the utilization of the products (both DKR and KR). Showcasing the applications or downstream uses of the PRODUCTS would enhance the paper.
3. Citation of Relevant Prior Work: The reviewer recommends citing a relevant paper (*Chem. Commun.*, 2022, 58, 4727) that reports on the kinetic resolution of non-point chiral substrates via amide N-alkylation with MBH-type carbonate. This addition would provide context and acknowledgment of related work in the field.

In addressing these points, the paper could strengthen its scientific rigor, transparency, and relevance to a broader readership.

Point-to-Point Responses to Reviewers' Comments (Manuscript ID: NCOMMS 23-48936-T)

Please take note that all the descriptive, positive comments of the reviewers are omitted, and only the reviewers' comments expressing their concerns/suggestions are listed below, which are followed by our responses. All the changes made in the revised manuscript as highlighted in yellow.

Revisions made in reply to Reviewer 1's comments:

- **Comment #1:** "Despite the authors and previous research labeling these distinctive tub-shaped eight-membered cyclic dilactams as planar chirality, this reviewer believes this uncommon type of chirality could also be termed inherent chirality. It is recommended to refer to a recent mini-review (Eur. J. Org. Chem. 2023, e202300738) which discusses this topic."
Our response: The above-mentioned paper has been properly cited in ref. 25, see the revised introduction section.
- **Comment #2:** "Regarding the kinetic resolution reaction scope, the authors have only tested racemic dianthranilides with mono-N-alkylated substrates. It would be insightful to know whether the authors have evaluated racemic dianthranilide substrates featuring other types of N-substitutions, including aryl, acyl, and the like."
Our response: As suggested, three new dianthranilides featuring phenyl (**4p**), Bz (**4q**) and Boc (**4r**) of N-substitutions have been prepared and subjected to the reaction, see the revised Figure 4. Notably, **4q** and **4r** have flexible conformations, and give DKR instead of DK results (**5q** & **5r**).
- **Comment #3:** "To elucidate the origins of the high stereoselectivity of these reactions, it would be beneficial to propose a plausible reaction mechanism and an asymmetric transition state."
Our response: Thank you for this insightful suggestion. A plausible reaction mechanism and an asymmetric transition state have been proposed accordingly to explain the origin of stereoselectivity, which is supported by preliminary DFT calculations, see and the revised manuscript (Figure 5 and corresponding text).

Revisions made in reply to Reviewer 2's comments:

- **Comment #1:** “This transformation represents a welcome contribution to the poor set of catalytic methodologies that produce planar chirality, but it focuses exclusively on a restricted framework and a particular transformation; the DKR substrate scope shows up to 25 different structures but, from a synthetic and applicability point of view, many of them are redundant.”

Our response: Thank you for acknowledging our work. As mentioned above, catalytic asymmetric synthesis of planar-chiral compounds is still in its infancy, mainly due to the lack of prototypes. So, recent progresses usually focus exclusively on a restricted framework and a particular transformation. For example, You and Zhou assembled planar-chiral ferrocenes via enantioselective C-H activation (ref. 41: *Nat. Synth.* 2022, 2, 49-57; ref. 44: *Nat. Chem.* 2023, 15, 815-823). Here, we report the catalytic asymmetric synthesis of planar-chiral dianthranilides via N-alkylation enabled (dynamic) kinetic resolution. Dianthranilide can serve as a new platform scaffold for studying planar chirality, which will arouse widespread interest among chemists and find broad applications in catalyst/ligand design and organic synthesis. Currently, we are trying to find and synthesize other planar-chiral frameworks, which will be reported in due course. For substrate scope, we believe that an appropriate amount of substrate is necessary. Only with sufficient specimens can we reach the correct conclusion, especially in the study of structure-activity relationship (SAR). Substrates and catalysts are 3D molecules, prior to testing, both the electron-nature and positions of the substituents may affect reactivity and selectivity. So here we have investigated the electronic properties and positions of substituents, and coincidentally, these data are very good, which in turn demonstrates a good versatility of this methodology.

- **Comment #2:** “To clarify the reaction mechanism, the authors performed several control experiments. The main concern is that the explanations and their representation are not easy to follow and are not presented in sufficient detail. Moreover, they do not imply, at any point, the role of the catalyst.”

Our response: It's a similar comment to **Reviewer 1's comment 3**. In revised manuscript, we have reorganized the control experiment section for ease of understanding, and proposed a detailed mechanism to explore the origin of stereoselectivity and the role of catalyst, which is supported by preliminary DFT calculations, see Figure 5 and the corresponding text.

- **Comment #3:** “On the other hand, the synthetic value of these planar chiral molecules as catalysts is exemplified in kinetic resolutions of secondary amines but the results provided are rather moderate.”

Our response: Thank you for this comment. To showcase the synthetic value of these planar chiral molecules, we tried to use them as acyl transfer reagent in kinetic resolutions of racemic amines. After further optimization, we found **1a'** could improve the resolution efficiency (from previous 50% *ee* to current 81% *ee*). Besides, have selected a few of the products (**3a**, **3f**, **3g**, **3h**, **3j**, **3l**, **3n** and **3o** for DKR, **5a**, **5e**, **5f**, **5g**, **5h**, **5n** and **5o** for KR) for the anticancer activity evaluation. The results show that compounds **3a**, **3o** and **5g** exhibited significant and broad anticancer potency, which also shows the synthetic value of these planar chiral molecules, see the revised Figure 6.

- **Comment #4:** “Finally, enantioenriched (+)-eupolyphagin is prepared. The evaluation given by the authors on page 2, of the synthetic route described earlier, is not too far from the one described by the authors themselves as a similar number of steps is needed and the enantioenriched product is obtained by kinetic resolution.”

Our response: Thank you for this comment. To demonstrate the practicality of this method, we prepared natural product eupolyphagin. Compared to Tan's work, although requiring a similar number of steps, our synthetic route obtained highly enantioenriched (+)-eupolyphagin without recrystallization. Besides, our protocol can provide both enantioenriched (+)-eupolyphagin and (-)-eupolyphagin at the same time (see the revised Figure 6C). Such an enantiodivergent strategy is of significant importance for natural product synthesis and drug discovery.

Revisions made in reply to Reviewer 3's comments:

- **Comment #1:** “Detailed Experimental Information in Supporting Information: The reviewer requests more comprehensive experimental details and results in the supporting information. Specifically, in case the substrate for (D)KR is a new compound, additional data for them, including chemical yield, product weight, physical properties, melting point, and elemental analysis or high-resolution mass spectrometry (HRMS) data, are needed. Notably, information on unsymmetrical substrates (1x and 1y) is entirely missing, and clarification is needed on whether these compounds were prepared using the same procedure as 1a. Additionally, for compounds 4k, 4l, 4m, 4n, and 4o, the reviewer highlights a lack of clarity on their synthesis process despite the provided chemical yield and spectral data. Preparation method for them, along with details on purification processes, should be provided.”

Our response: The SI has been carefully re-checked, and detailed experimental procedures and results have been updated accordingly. For new compounds, chemical yield, product weight, physical properties, melting point, and high-resolution mass spectrometry (HRMS) data, have been provided. Detailed information on unsymmetrical

substrates (**1x** and **1y**) has been added. Preparation method along with details on purification processes for compounds **4k**, **4l**, **4m**, **4n**, and **4o** have been reported as suggested.

- **Comment #2:** “Demonstration of Product Usage: The reviewer notes that while the paper successfully demonstrates the use of non-reacted, recovered starting compounds for KR, it falls short in illustrating the utilization of the products (both DKR and KR). Showcasing the applications or downstream uses of the PRODUCTS would enhance the paper.”

Our response: Thank you for this good suggestion. To demonstrate the downstream uses of the PRODUCTS, we have selected a few of the products (**3a**, **3f**, **3g**, **3h**, **3j**, **3l**, **3n** and **3o** for DKR, **5a**, **5e**, **5f**, **5g**, **5h**, **5n** and **5o** for KR) for the anticancer activity evaluation on cell viability via the CTG assay for A2780 (ovarian cancer cells), HeLa (cervical carcinoma cells), HT-29 (colon cancer cells), LoVo (colon cancer cells), MV-4-11 (acute monocytic leukemia cells), and U87-MG (astrocytoma cells) human cancer cell lines. The results show that compounds **3a** (A2780 cells, IC₅₀ = 13.89 μM; HeLa cells, IC₅₀ = 37.28 μM; HT-29 cells, IC₅₀ = 28.41 μM; LoVo cells, IC₅₀ = 36.25 μM; MV-4-11 cells, IC₅₀ = 21.00 μM; U87-MG cells, IC₅₀ = 47.68 μM), **3o** (A2780 cells, IC₅₀ = 16.24 μM; HeLa cells, IC₅₀ = 61.51 μM; HT-29 cells, IC₅₀ = 54.07 μM; LoVo cells, IC₅₀ = 53.60 μM; MV-4-11 cells, IC₅₀ = 38.79 μM; U87-MG cells, IC₅₀ = 109.33 μM), and **5g** (A2780 cells, IC₅₀ = 10.58 μM; HeLa cells, IC₅₀ = 16.80 μM; HT-29 cells, IC₅₀ = 15.06 μM; LoVo cells, IC₅₀ = 14.75 μM; MV-4-11 cells, IC₅₀ = 14.27 μM; U87-MG cells, IC₅₀ = 25.43 μM) exhibited significant and broad anticancer potency. These results show that these PRODUCTS can become potential anticancer lead compounds. A further structure–activity relationship study is ongoing in our laboratory and will be reported in due course.

- **Comment #3:** “Citation of Relevant Prior Work: The reviewer recommends citing a relevant paper (Chem. Commun., 2022, 58, 4727) that reports on the kinetic resolution of non-point chiral substrates via amide N-alkylation with MBH-type carbonate. This addition would provide context and acknowledgment of related work in the field.”

Our response: Thank you for this good suggestion. The above-mentioned paper and some other relevant works on amide N-alkylation have been properly cited in ref. 60-63, see the revised manuscript.

REVIEWER COMMENTS

Reviewer #1 (Remarks to the Author):

Since all my concerns have been well addressed, I recommend the acceptance of this manuscript in Nature Communications.

However, it is noteworthy that during the revision of this manuscript, two other examples of asymmetric synthesis of conformationally rigid chiral medium-sized cyclic molecules has been published, which are recommended to be cited. (10.1002/anie.202319289, 10.1016/j.cheecat.2023.100827).

Reviewer #2 (Remarks to the Author):

In response to the authors' letter, this referee still has reservations about the overall interest of the specific dianthranilide framework, despite the authors' attempts to justify it through comparisons with other works in the journal and the potential interest it may generate within the organic chemistry community.

Regarding the reaction mechanism, the DFT calculations provided in this new version are not informative. Although the catalyst is involved, as suggested, they do not provide activation energy values for the transition states, which are essential to explain the observed stereoselectivity. Consequently, this new section does not provide what was requested to be explained in more detail.

Reviewer #3 (Remarks to the Author):

This revised version adequately addresses the comments from this reviewer.

Reviewer #4 (Remarks to the Author):

In this work, Mei et.al. reported the first catalytic asymmetric synthesis of planar-chiral dianthranilides enabling by cinchona alkaloid-catalyzed (dynamic) kinetic resolution. Under mild conditions, various C₂- or C₁-symmetric planar-chiral dianthranilides were readily prepared in high yields with excellent enantioselectivities. Several planar-chiral dianthranilides prepared in this work were shown to have good anti-cancer activity and may have the potential to become new anticancer lead compounds. After carefully reading the whole manuscript, I felt that this paper could be published in Nature Communication after minor revision.

For the substrate scope of dianthranilides in Figure 2, the author mentioned that "The reaction was applicable to a brand range of symmetrically substituted dianthranilides, leading to the formation of dialkylated products 3g-t in generally excellent yields and enantioselectivities." Substituents such as -CF₃, -NO₂, -OCF₃ and tert-butyl group should also be studied to evaluate the broad substrate scope of dianthranilides.

Point-to-Point Responses to Reviewers' Comments (Manuscript ID: NCOMMS-23-48936A)

Please take note that all the descriptive, positive comments of the reviewers are omitted, and only the reviewers' comments expressing their concerns/suggestions are listed below, which are followed by our responses. All the changes made in the revised manuscript as highlighted in yellow.

Revisions made in reply to Reviewer 1's comments:

- **Comment #1:** "Since all my concerns have been well addressed, I recommend the acceptance of this manuscript in Nature Communications. However, it is noteworthy that during the revision of this manuscript, two other examples of asymmetric synthesis of conformationally rigid chiral medium-sized cyclic molecules has been published, which are recommended to be cited. (10.1002/anie.202319289, 10.1016/j.checat.2023.100827)."
Our response: The above-mentioned papers have been properly cited in ref. 55-56, see the revised manuscript.

Revisions made in reply to Reviewer 2's comments:

- **Comment #1:** "Regarding the reaction mechanism, the DFT calculations provided in this new version are not informative. Although the catalyst is involved, as suggested, they do not provide activation energy values for the transition states, which are essential to explain the observed stereoselectivity. Consequently, this new section does not provide what was requested to be explained in more detail."
Our response: Thank you for this comment. We have performed a detailed DFT calculation of transition states to explore the stereoselectivity, and the detailed results (including the activation energy values) have been added to the revised SI.

Revisions made in reply to Reviewer 3's comments:

- **Comment #1:** "This revised version adequately addresses the comments from this reviewer."
Our response: We highly appreciate your previous valuable comments, which are very helpful for improving our manuscript.

Revisions made in reply to Reviewer 4's comments:

- **Comment #1:** "For the substrate scope of dianthranilides in Figure 2, the author mentioned that "The reaction was applicable to a broad range of symmetrically substituted dianthranilides, leading to the formation of dialkylated products 3g-t in generally excellent yields and enantioselectivities." Substituents such as -CF₃, -NO₂, -OCF₃ and tert-butyl group should also be studied to evaluate the broad substrate scope of dianthranilides."

Our response: As suggested, some additional dianthranilides featuring -CF₃, -NO₂, -CN and -^tBu substitutions have been prepared and subjected to the reaction. The starting materials used for preparing the substrate with the -OCF₃ group are not readily available, and instead the -CN substituted dianthranilide has been synthesized. Among them, -^tBu (**3k**) and -CF₃ (**3n**) substitutions give the consistent good results, see the revised Figure 2. However, dianthranilides bearing -NO₂ and -CN substitutions exhibit strong polarities and are difficult to dissolve in common organic solvents (DCM, CH₃CN, toluene, and THF). So, these two substrates afford the results of no reaction, see the revised SI for details.

REVIEWER COMMENTS

Reviewer #2 (Remarks to the Author):

While this second revised version of the manuscript has made progress in addressing suggestions regarding the theoretical study of the reaction mechanism, I find that the section dedicated to explaining the observed stereoselectivity does not meet the quality standards expected by Nat. Commun. Although the authors have included energy values for the transition states in the supporting information, it is unclear why these data have not been used to thoroughly discuss the stereochemical outcome of the reaction in the text. Furthermore, no indications about alternative interaction modes are provided to confirm that the proposed model is the most plausible one.

Reviewer #5 (Remarks to the Author):

In this manuscript, Mei and co-workers report the first catalytic asymmetric synthesis of planar-chiral dianthranilides via (dynamic) kinetic resolution. A wide range of tub-shaped eight-membered cyclic dilactams were obtained in good yields with excellent enantioselectivities. The broad scope of the reaction, the gram-scale synthesis, the enantiodivergent synthesis of natural product and the antitumor activity evaluation were well performed. As opposite to the well-studied central, axial, and helical chirality, less attention has been paid to planar chirality due to the difficulty in synthesis and the lack of prototype. Consequently, the catalytic asymmetric synthesis of planar-chiral dianthranilides is of outstanding scientific quality and originality. The findings described in this manuscript will be of great interest to those in the field of synthetic and medicinal chemistry as well as asymmetric catalysis. Nevertheless, there are some small issues on the discussion of DFT calculations. To meet the high standard of Nature Communications, this manuscript needs minor revisions.

1. The DFT calculations well supported the proposed mechanism and provided insights into the and origins of the enantioselectivity. I have checked the structures provided in Supporting information, the transition states obtained with DFT calculations are reasonable. However, the authors should check carefully for the way to draw the structures of transition state TS-S and TS-R (also, TS-1 TS-2 in Figure 5 and Figure S1), especially for the chirality of the substrates in these transition states.
 2. Page S213 in Supporting information: the Gibbs energies for TS-S ($\Delta G^\ddagger = 15.3$ kcal/mol) and TS-R ($\Delta G^\ddagger = 17.8$ kcal/mol) are different with those in Figure S1 (TS-R ($\Delta G^\ddagger = 15.3$ kcal/mol) and TS-S ($\Delta G^\ddagger = 17.8$ kcal/mol)). Please correct them.
 3. Please provide the details for enantiomeric excess calculation with Boltzmann distribution, at least the equation should be provided in Supporting information.
 4. Some unreadable symbol appeared in the NCI plot in Figure S3, please correct them.
 5. In my understand, the structure for TS-S and TS-R are the same with TS-1 and TS-2, right? If yes, please use the unified name for the same structure. Also, the way to draw the same structure should keep the same in the same manuscript to avoid confusing.
- Overall, the manuscript is well written, and SI is of high quality. I strongly support the publication of this manuscript in Nature Communications.

Point-to-Point Responses to Reviewers' Comments (Manuscript ID: NCOMMS-23-48936B)

Please take note that all the descriptive, positive comments of the reviewers are omitted, and only the reviewers' comments expressing their concerns/suggestions are listed below, which are followed by our responses. All the changes made in the revised manuscript as highlighted in yellow.

Revisions made in reply to Reviewer 5's comments:

- **Comment #1:** "The DFT calculations well supported the proposed mechanism and provided insights into the and origins of the enantioselectivity. I have checked the structures provided in Supporting information, the transition states obtained with DFT calculations are reasonable. However, the authors should check carefully for the way to draw the structures of transition state TS-S and TS-R (also, TS-1 TS-2 in Figure 5 and Figure S1), especially for the chirality of the substrates in these transition states."
Our response: The structures of transition state and all other DFT calculations have been carefully checked accordingly. Appropriate corrections have been made to the discovered errors, please refer to the revised SI.
- **Comment #2:** "Page S213 in Supporting information: the Gibbs energies for TS-S ($\Delta G^\ddagger = 15.3$ kcal/mol) and TS-R ($\Delta G^\ddagger = 17.8$ kcal/mol) are different with those in Figure S1 (TS-R ($\Delta G^\ddagger = 15.3$ kcal/mol) and TS-S ($\Delta G^\ddagger = 17.8$ kcal/mol)). Please correct them."
Our response: It has been corrected accordingly.
- **Comment #3:** "Please provide the details for enantiomeric excess calculation with Boltzmann distribution, at least the equation should be provided in Supporting information."
Our response: The details for enantiomeric excess calculation with Boltzmann distribution have been provided accordingly, see the revised SI.
- **Comment #4:** "Some unreadable symbol appeared in the NCI plot in Figure S3, please correct them."
Our response: It has been revised accordingly.
- **Comment #5:** "In my understand, the structure for TS-S and TS-R are the same with TS-1 and TS-2, right? If yes, please use the unified name for the same structure. Also, the way

to draw the same structure should keep the same in the same manuscript to avoid confusing.”

Our response: Yes, the structures for TS-S and TS-R are the same with TS-1 and TS-2. In our revised SI, they are unified and drawn in the same way accordingly.

REVIEWERS' COMMENTS

Reviewer #5 (Remarks to the Author):

The authors have addressed all the issues, I would like to recommend this paper publish in Nature Communications.